# Domain Specific and Cross Domain Associations between PASS Cognitive Processes and Academic Achievement

**DOI:** 10.3390/bs13100824

**Published:** 2023-10-07

**Authors:** Sergios C. Sergiou, George K. Georgiou, Charalambos Y. Charalambous

**Affiliations:** 1Department of Primary Education, University of Cyprus, 4071 Nicosia, Cyprus; cycharal@ucy.ac.cy; 2Department of Educational Psychology, University of Alberta, Edmonton, AB T6G 2G5, Canada; georgiou@ualberta.ca

**Keywords:** attention, Greek, intelligence, mathematics, reading, PASS processes, planning

## Abstract

The purpose of this study was to examine the role of intelligence—operationalized in terms of Planning, Attention, Simultaneous, and Successive (PASS) processing skills—in reading and mathematics. Two hundred and forty-two Grade 6 Greek-speaking students (114 boys and 128 girls, *M*_age_ = 135.65 months, *SD* = 4.12 months) were assessed on PASS processes, speed of processing (Visual Matching), reading (Wordchains and CBM-Maze), and mathematics (Mathematics Achievement Test and Mathematics Reasoning Test). The results of the hierarchical regression analyses showed that, after controlling for family’s socioeconomic status and speed of processing, Attention and Successive processing predicted reading and Planning and Simultaneous processing predicted mathematics. Taken together, these findings suggest that different PASS processes may account for individual differences in reading and mathematics.

## 1. Introduction

Several studies over the last two decades have examined the domain-specific and cross-domain effects of different cognitive skills on reading and mathematics, e.g., [1,2,3,4]. Identifying cross- and within-domain predictors is important because it has significant implications for cognitive developmental theories and for practice. Cognitive skills such as phonological awareness, morphological awareness, and orthographic knowledge have been recognized as crucial for reading. Visual–spatial memory and speed of processing have been identified as crucial for mathematics. Finally, skills such as rapid automatized naming and executive functioning have been found to play an important role in both reading and mathematics. Despite the fact that we now have a much better understanding of the domain-specific and cross-domain effects of different cognitive skills on reading and mathematics, much less is known about the role of intelligence. Thus, the purpose of this study was to explore the role of intelligence—operationalized in terms of Planning, Attention, Simultaneous, and Successive (PASS) neurocognitive processes—in reading and mathematics. 

### 1.1. PASS Theory of Intelligence

To date, several studies have shown that intelligence is related to academic achievement. In a recent meta-analysis, Lozano-Blasco et al. [5] estimated the size of this relation to be *r* = 0.36. However, some researchers have raised concerns about the way intelligence has traditionally been operationalized, e.g., [6,7]. More specifically, they indicated that some popular batteries of intelligence (e.g., WISC) include tasks such as Mathematics and Vocabulary that are very similar to achievement tests and thus measure more students’ “knowing” than “thinking”, which should be the target of intelligence testing. In response to this criticism, in 1994, Das and colleagues proposed the Planning, Attention, Simultaneous, and Successive processing (PASS) theory of intelligence and developed the Cognitive Assessment System (CAS) battery to operationalize it. 

The PASS theory of intelligence draws on Luria’s work on brain organization and functioning. According to Luria [8], human cognition consists of three functional units that support four neurocognitive processes (Planning, Attention, Simultaneous, and Successive (PASS) processing). The first functional unit is Attention. Attention is the ability to demonstrate selective, sustained, and effortful activity over time and resistance to distractions. The second functional unit—an information processing unit—consists of two sub-processes, namely Simultaneous and Successive processing. Simultaneous processing allows individuals to integrate stimuli into interrelated groups usually found in tasks with strong visual–spatial demands. In turn, Successive processing allows individuals to process information in serial order, including the perception of stimuli in sequence and the linear execution of steps in a given task. Finally, the third functional unit, Planning, allows individuals to develop or select strategies to solve a problem and to monitor progress in solving the problem. 

Das and colleagues [9] theorized that Successive processing influences word recognition through the effects of phonological recoding (i.e., sounding out) and Simultaneous processing influences word recognition through the effects of orthographic knowledge (i.e., the formation, storing, and processing of whole word representations). Planning and Attention support Simultaneous and Successive processing and enable the deployment of phonological recoding and orthographic knowledge. Previous studies with typically developing children, e.g., [10,11], as well as special populations, e.g., [12,13], have confirmed these predictions. Intervention studies have also shown that training children in Simultaneous and Successive processing resulted in a significant improvement in their word recognition, e.g., [14,15].

Nevertheless, most of the aforementioned studies included children in early grades and did not examine for possible connections with reading comprehension. Given that reading comprehension involves developing a coherent representation of a text that requires efficient integration of information across the text, one would expect Simultaneous processing to be a significant predictor of reading comprehension. Planning should also play a significant role because to succeed in comprehension, individuals need to develop a plan on how to approach a passage and to monitor their comprehension as they read. The few studies that examined the role of PASS processes in reading comprehension have produced mixed findings. For example, in a study with Greek-speaking adolescent students, Kendeou et al. [16] showed that Planning and Attention had a direct effect on reading comprehension (operationalized with CBM-Maze) and that Simultaneous and Successive processing predicted reading comprehension through the effects of word reading. In contrast, working with a group of English-speaking university students, Georgiou and Das [17] showed that none of the PASS processes had a direct effect on reading comprehension. Attention, Simultaneous, and Successive processing predicted word- and text-reading fluency, which, in turn, predicted reading comprehension. Thus, more research is needed on the role of PASS processes in reading comprehension. 

Recently, Deaño et al. [18] have also proposed specific connections between the PASS processes and mathematics. According to Deaño et al. [18], calculations rely on Planning/Executive Functioning because children must come up with a strategy on how to solve a specific operation. In turn, problem solving relies on Simultaneous processing. Simultaneous processing involves logical–grammatical relations. Nonverbal matrix-type tests and verbal Simultaneous tests are both used to assess logical–grammatical relations in word problems. Das and Janzen [19] also argued that seeing similarities between two problems and transferring procedures learned from one problem to another, falls under the scope of Simultaneous processing. Previous studies with typically developing children, e.g., [20,21,22], as well as children with mathematics difficulties, e.g., [23,24], have confirmed these predictions.

With the exception of the CAS standardization study [25], only two independent studies have examined the role of PASS processes in both reading and mathematics within the same study [20,26]. Working with a group of typically developing Kindergarten children, Georgiou et al. [20] showed that Planning and Successive processing were unique predictors of word recognition and reading fluency in Grade 1 and that none of the PASS processes were predictive of mathematics, after controlling for the effects of phonological awareness and visual–spatial memory. In turn, Dunn et al. [26] found that Planning and Successive processing were significant predictors of Broad Reading (Broad Reading is a cluster score derived from scores in the Letter–Word Identification, Reading Fluency, and Passage Comprehension sections of the Woodcock–Johnson test of achievement) in a sample of intellectually gifted students in Grades 4 to 6 and that Planning and Simultaneous processing were unique predictors of Broad Mathematics (Broad Mathematics is a cluster score derived from the Calculation, Math Fluency, and Applied Problems sections). Even though the two studies showed that Planning and Successive processing were unique predictors of reading, their findings diverged in terms of the predictors of mathematics. Clearly, more studies are needed to examine the domain-specific and cross-domain effects of PASS processes.

### 1.2. The Present Study

The purpose of this study was to examine the role of PASS cognitive processes in reading and mathematics in a sample of typically developing Greek-speaking children. Based on the proposed theoretical links between PASS processes and reading/mathematics [9,18,19], we expected that Simultaneous and Successive processing would predict reading and that Planning and Simultaneous processing would predict mathematics. In addition, because of the established connection between Attention Deficit Hyperactivity Disorder (ADHD) and reading, e.g., [27,28], as well as between inhibition and reading, e.g., [29,30], we expected that Attention would also predict reading.

It is worth noting that in this study, we controlled for the effects of two very important variables, namely family’s socioeconomic status (SES) and speed of processing. In regard to the former, several studies have shown that it is a significant correlate of academic achievement (see [31,32], for evidence from meta-analyses) and may partly account for individual differences in PASS processes. In regard to the latter, controlling for speed of processing was necessary for two reasons: First, because some of the PASS (e.g., Planned Codes and Expressive Attention) and reading (e.g., Wordchains) tasks in our study are speeded, we wanted to control for the effects of speed of processing in order to capture the “true” effect of the PASS processes. Second, according to Best et al. [33], the Planning and Attention tasks from CAS represent “complex” and “simple” executive functioning, respectively. Some researchers have argued that the effects of speed of processing must be controlled before examining the effects of executive functioning on academic achievement, e.g., [34,35].

## 2. Methods

### 2.1. Participants

Letters of information describing our study were sent to the families of 273 Grade 6 students attending 16 public elementary schools (5 urban, 7 suburban, and 1 rural) in Cyprus. After excluding students who did not receive parental consent (*n* = 21) and students who immigrated recently in Cyprus and could not communicate well in Greek (*n* = 10), our sample comprised 242 students (114 boys, 128 girls, *M*_age_ = 135.65 months, *SD* = 4.12 months). All students were native speakers of Greek, and none were diagnosed with any intellectual, emotional, or sensory disabilities (based on school records). The study was conducted in accordance with the Declaration of Helsinki and the protocol was approved by the Cyprus Ethics Committee (National Institutional Review Board approval number 141690).

### 2.2. Materials

Family’s Socioeconomic Status (SES). A family’s SES was assessed with two items. First, we asked students to indicate how many children’s books they had at home by choosing one of five options (1 = 0–10 books; 2 = 11–25 books; 3 = 26–100 books; 4 = 101–200 books; 5 = more than 200 books). Next, following Kyriakides et al. [36], we collected information on parental occupation, which was coded according to five categories (0 = unemployed; 1 = machine operator, hospitality staff, assistant, laborer, and related worker; 2 = tradesperson, clerk and skilled officer, and sales and service staff; 3 = other business manager, arts/media/sportsperson, and associate professional; 4 = senior manager in a large business organization, government administration, and qualified professional). Parental occupation score was the average score of mother’s and father’s occupations.

PASS Cognitive Processes. To assess the PASS cognitive processes, we used the Cognitive Assessment System (CAS)-2: Brief [37]; see also [38] for a validation study of CAS-2: Brief in Greek. Below is a description of the measures in CAS-2: Brief.

Planning. Planned Codes was used to assess Planning. At the top of each of the six items that this test comprised, there was a legend, which indicates how the numbers relate to specific combinations of O’s and X’s (e.g., 1 = OX; 2 = XX; 3 = OO; 4 = OX). Children were instructed to fill in, as quickly as possible using any strategy, 32 empty boxes with a combination of O’s and X’s (e.g., 1 = OX; 2 = XX; 3 = OO; 4 = OX). Children were given 60 s to fill in as many empty boxes as possible. Each item contained a different arrangement of the numbered boxes and O/X combinations. The time and number correct for each page were recorded and combined to obtain a ratio score. The ratio score was then converted to a subtest scaled score. Cronbach’s alpha in our sample was 0.81.

Attention. Expressive Attention, a transparent adaptation of the Stroop task [39], was used to assess Attention. It contains three items of increasing difficulty, and children were given 180 s to complete each item. In the first item, children were asked to read, as fast as possible, the names of colors written in their respective colors (i.e., Blue, Yellow, Green, and Red) arranged in eight rows of five. In the second item, children were asked to name, as fast as possible, the colors of a sequence of rectangles. In the third item, the names of colors were printed in a different color from the named color (e.g., the word Red may appear in blue) and children were asked to name the color of the ink, not to read the text. The time and number of correct answers in the last item were recorded and combined to obtain a ratio score. The ratio score was then converted to a scaled score. Cronbach’s alpha in our sample was 0.84.

Simultaneous processing. Nonverbal matrices were used to assess Simultaneous processing. This task is similar to Raven’s Progressive Matrices [40]. The nonverbal matrices consisted of 44 items that present a variety of shapes and geometric designs, each having a missing piece. The shapes and geometric designs were interrelated through spatial or logical organization. For each item, the children were required to decode the relationships and to choose from a list of six possible answers to complete the picture. The task was discontinued after four consecutive errors. The subtest score was the total number of correct answers. Cronbach’s alpha in our sample was 0.85.

Successive processing. Successive Digits was used to assess Successive processing. This task is similar to Digit Span Forward [41]. It contains 28 items of different digit sequences, varying in length from 2 to 9 digits. Children were first asked to listen carefully to the tester saying out loud a string of digits (e.g., 2, 9, 5) and then were asked to repeat the string of digits in the same order. The task was discontinued after four consecutive errors. The subtest score was the total number of correctly repeated strings of digits. Cronbach’s alpha in our sample was 0.79.

Speed of processing. To assess speed of processing, we administered the Visual Matching task from the Woodcock–Johnson Tests of Cognitive Ability battery [42]. The task consisted of 60 rows of numbers with six numbers in each row. Two of the numbers in each row were the same (e.g., 8, 9, 5, 2, 9, 7) and the children were asked to circle the identical numbers in each row as fast as possible. Children completed four practice items prior to timed testing. A participant’s score was the total number of rows completed correctly within a 3 min time limit. Woodcock et al. [43] reported test–retest reliability for Visual Matching to be 0.87 for 7- to 11-year-olds.

Reading. To assess reading, we administered two tasks: Wordchains [44] and CBM-Maze [45]. The Wordchains task follows a similar format as the Test of Silent Word Reading Fluency [46]. Children were asked to put slashes to separate words that are written without any spaces between them (e.g., μέσατώραφως → μέσα/τώρα/φως). The test had a total of 15 rows of words of increasing length and a participant’s score was the number of correctly separated words (max = 180) within a 1 min time limit. Cronbach’s alpha reliability in our sample was 0.89. The CBM-Maze task was developed in Greek following the principles of the CBM-Maze task in English; see [47,48]. The children were asked to read a short passage (295 words) in which every seventh word was replaced with three options and to circle the option that was correctly completing the meaning of each sentence. A participant’s score was the number of correct answers minus the number of incorrect answers. The children were given three minutes to complete the task. Cronbach’s alpha reliability in our sample was 0.90. In addition, CBM-Maze correlated *r* = 0.65 with Wordchains in our study.

Mathematics. To assess mathematics, we administered two tasks: the Mathematics Achievement Test (MAT) [36] and the Mathematics Reasoning Test (MRT) [49]. MAT includes 13 items (with sub-items) and measures children’s performance in basic mathematics notions and operations (e.g., comparing and operating on numbers, finding the perimeter and area of given shapes) as well as in problem solving. Cronbach’s alpha reliability in our study was 0.91. In turn, MRT includes 17 items (with sub-items) and measures students’ performance in reasoning with respect to functional thinking and general arithmetic. Cronbach’s alpha reliability in our study was 0.92. Both mathematics tests have been used in previous studies in Cyprus with children of similar age as the participants of our study and have shown very good psychometric properties [36,49]. MAT correlated *r* = 0.74 with MRT in our study.

### 2.3. Procedure

Children were assessed in their schools by trained research assistants. Testing was completed in three phases. During the first phase, the children were individually administered the CAS-2: Brief cognitive processing measures and the Visual Matching test. This phase lasted approximately 35 min. During the second phase, children were administered the Wordchains, CBM-Maze, and the MAT task. Finally, during the third phase, they were administered the MRT task. Phases 2 and 3 lasted approximately 40 min each. All children were assessed in the three phases in the same order, and the time difference between each phase of testing was a week.

### 2.4. Data Analysis

Descriptive statistics and Pearson’s correlations were initially calculated to provide an overview of the study variables and to examine the relation between the students’ academic achievement and the PASS cognitive processes. Next, we conducted two sets of hierarchical regression analyses to examine the contribution of the four PASS processes to reading (Wordchains and CBM-Maze) and mathematics (MAT and MRT), separately for each outcome. In the first set of hierarchical regression analyses, at Step 1, we entered the regression equation parents’ occupation and number of children’s book at home. At Step 2, we entered the speed of processing. Finally, at Step 3, we entered each of the four PASS processes separately. In the second set of hierarchical regression analyses, we controlled for the number of children’s books at home and parental occupation (Step 1); then, entered the speed of processing at Step 2; and finally, entered the four PASS processes as a block at Step 3. For each analysis conducted, an alpha level of 0.05 was used for judging statistical significance. Statistical analyses were conducted using SPSS Statistics 24.

## 3. Results

### 3.1. Descriptive Statistics and Correlations

Table 1 presents the descriptive statistics of our measures. An inspection of the Q-Q plots revealed that our measures were normally distributed and the Shapiro–Wilk tests were all non-significant [50]. Table 2 presents the Pearson product moment correlations. The correlations between each PASS subscale and students’ performance ranged from *r* = 0.22 to *r* = 0.54 for mathematics and *r* = 0.28 to *r* = 0.41 for reading, with Successive processing having the lowest correlation with both mathematics tasks and Planning having the highest. Furthermore, the correlations between the speed of processing and reading and mathematics tasks were moderate to strong (*r* = 0.33 up to *r* = 0.58), with the speed of processing having the lowest correlation with MRT and the highest with CBM-Maze. The two indicators of students’ SES were more strongly correlated with the mathematics outcomes (*r* = 0.45 to *r* = 0.52) than the reading outcomes (*r* = 0.20 to *r* = 0.26).

### 3.2. Hierarchical Regression Analyses

The results of the hierarchical regression analyses are presented in Table 3. In Model 1, the two SES indicators had a significant contribution to both reading and mathematics, explaining 7% of the variance in reading and 25% to 31% of the variance in mathematics. Speed of processing (entered at Step 2) had a significant contribution to students’ performance in all four outcomes, accounting for an additional 13% to 29% of the variance in reading and 5% to 10% of the variance in mathematics. Finally, with the exception of Attention, which did not significantly predict MRT, the rest of the PASS processes (entered interchangeably at Step 3) accounted for a unique amount of variance in all four outcomes. Planning (β = 0.12 to β = 0.19, *p* < 0.05), Attention (β = 0.20 to β = 0.21, *p* < 0.01), Simultaneous processing (β = 0.18, *p* < 0.001), and Successive processing (β = 0.19 to β = 0.26, *p* < 0.001) significantly contributed to students’ performance in both reading tests, explaining 1% to 9% of unique variance. For both MAT and MRT, Planning (β = 0.33 to β = 0.35, *p* < 0.001), Simultaneous processing (β = 0.28, *p* < 0.001), and Successive processing (β = 0.17 to β = 0.20, *p* < 0.01) significantly predicted students’ performance, and Attention also predicted students’ performance in MAT (β = 0.17 *p* < 0.01). Each of the four PASS processes explained an additional 2% to 10% of the variance in the mathematics outcomes.

In Model 2, the four PASS processes (entered as a block at Step 3) collectively accounted for 8–9% of the variance in reading and 13–14% of the variance in mathematics. Interestingly, of the four PASS processes, only Attention and Successive processing remained significant predictors of reading (Attention: β_Wordchains_ = 0.14, *p* < 0.05, β_CBZ-Maze_ = 0.14, *p* < 0.05; Successive processing: β_Wordchains_ = 0.22, *p* < 0.01, β_CBZ-Maze_ = 0.13, *p* < 0.05). In turn, when predicting the two mathematics outcomes, only Planning and Simultaneous processing remained significant predictors (Planning: β_MAT_ = 0.27, *p* < 0.001, β_MRT_ = 0.26, *p* < 0.001; Simultaneous processing: β_MAT_ = 0.14, *p* < 0.001, β_MRT_ = 0.18, *p* < 0.001).

## 4. Discussion

The purpose of this study was to examine the role of PASS cognitive processes in both reading and mathematics in a sample of Grade 6 Greek-speaking students. We hypothesized that Attention, Successive, and Simultaneous processing would predict reading and that Planning and Simultaneous Processing would predict mathematics. Our hypotheses were partly confirmed. After controlling for SES and speed of processing, there was a clear division in the contribution of the four PASS processes. Of the two auxiliary cognitive processes, Attention predicted reading and Planning predicted mathematics. Both of these findings are in line with those of previous studies, e.g., [24,25,26,51,52], as well as with the broader literature on executive functioning. Planning, the pinnacle of executive functioning, involves developing a strategy to solve a problem and evaluating the progress in reaching one’s goal. As Das and Misra [53] pointed out, Planning is critical in problem solving and that is why it is a strong correlate of mathematics achievement. In line with our findings, Clark et al. [54] showed that Planning (measured with the Tower of Hanoi task when the children were four years old) was a significant predictor of children’s mathematics fluency (but not of reading comprehension) at the age of 6. Notably, children who failed the initial levels of the Tower of Hanoi task at the age of 4 showed a significant decrease in their mathematics fluency at the age of 6. In contrast, progressively higher achievement in the Tower of Hanoi task was related to a five-point increase in performance in mathematics fluency. Evidence in support of the role of Planning in mathematics achievement also comes from intervention studies [55,56]. For example, Naglieri and Johnson [55] have also shown that training in Planning improves children mathematics fluency.

In turn, Expressive Attention, the measure used to operationalize Attention in CAS:2-Brief, is similar to the Stroop task that is one of the most commonly used measures of inhibition (one of the components of executive functioning [57]). Several studies have shown that inhibition is a significant predictor of word reading and reading comprehension, e.g., [29,30,58,59]. For example, Borella and de Ribaupierre [59] showed that the ability to inhibit irrelevant information was a significant predictor of reading comprehension even in the presence of working memory and speed of processing. They further suggested that preventing irrelevant information from entering and cluttering working memory is important for comprehension when there is a significant memory load to cope with in the process of retrieving information to answer comprehension questions. In CBM-Maze, children must also inhibit the other choices in every seventh word in order to accurately construct a coherent representation of the text. In Wordchains, children must also inhibit the activation of competing orthographic neighbors (i.e., words that differ with only one letter from each other) in order to put the slash in the right place.

Of the two information processing skills, Successive processing predicted reading and Simultaneous processing predicted mathematics. The significant contribution of Successive processing to reading was expected given that Wordchains requires some phonological recoding until children discover the boundaries of a given word and that the options in CBM-Maze must be processed serially until a student makes a decision. In addition, Successive Digits (the task used here to operationalize Successive processing) is similar to Digit Span Forward, which is one of the most common measures of the phonological loop (one of the three components of working memory; see [60]). Several studies have shown that the phonological loop relates more strongly with reading than with mathematics, e.g., [61,62].

However, we also expected Simultaneous processing to predict reading because success in Wordchains would require some level of whole word recognition (see [63]) that is supported by Simultaneous processing [9] and success in CBM-Maze would require processing of logical–grammatical relations in the text in order to select the right words and to develop a coherent representation of the text. This unexpected finding may relate to the fact that we operationalized Simultaneous processing with a non-verbal task (i.e., Simultaneous Matrices) and not with a verbal task (i.e., Verbal–Spatial Relations) that would probably be more closely connected to reading outcomes. This may also explain why Simultaneous processing was a significant predictor of the two mathematics tasks that did not contain too much verbal information.

Some limitations of the present study are worth reporting. First, our study is correlational and any significant relations do not mean causation. Second, we used single measures to operationalize the PASS processes. Even though we were hoping to administer the whole CAS-2 battery, we were constrained by the time allowed by the school authorities to carry out our research. Third, even though CBM-Maze has been used in several studies as a reading comprehension measure see, e.g., [16], it does not require higher-level comprehension skills. As Das and Georgiou [52] have shown, the choice of comprehension measure may influence the role of Planning, which may explain why Planning did not predict CBM-Maze in our study. A future study should replicate our findings using higher-level comprehension tasks. Finally, our study was conducted with Grade 6 students and we do not know if our findings generalize to early grades.

Even though the theoretical models connecting PASS processes to reading [9] and mathematics [18] predicted some overlap in the effects of PASS processes on reading and mathematics (particularly in relation to Simultaneous processing), our findings showed that, to the extent there is an overlap, this pertains to Planning (it predicted CBM-Maze and the two mathematics tasks) and Successive processing (it predicted the two reading tasks and MAT). Overall, these findings suggest that the relation of PASS processes with reading and mathematics is not as simple as initially thought and needs to consider factors such as the nature of the reading and mathematics tasks, and the age of the participants when these relations are under examination.

## Figures and Tables

**Table 1 behavsci-13-00824-t001:** Descriptive statistics for the measures used in the study.

	Min.	Max.	Mean	SD	Skewness	Kurtosis
1. Age	130	155	135.65	4.12	0.89	1.65
2. Number of Books at Home ^a^	1	5	2.5	1.11	0.32	−0.72
3. Parental Occupation ^b^	0	4	2.5	1.17	0.25	−0.59
4. Speed of processing	7	34	22.73	5.53	−0.63	−0.26
5. Planning	88	141	113.05	9.77	−0.07	0.09
6. Attention	75	152	108.40	10.59	0.17	1.24
7. Simultaneous processing	77	115	93.24	9.10	0.32	−0.71
8. Successive processing	77	121	96.50	9.05	0.45	−0.91
9. Wordchains	4	36	14.74	5.07	0.60	0.38
10. CBM-Maze	8	46	24.88	7.86	0.21	−0.35
11. Mathematics Achievement Test	2	43	27.81	9.83	0.15	−0.97
12. Mathematics Reasoning Test	0	46	20.60	11.45	−0.56	−0.50

Note: ^a^ 1: 0–10 books; 2: 11–25 books; 3: 26–100 books; 4: 101–200 books; 5: more than 200 books. ^b^ 0: unemployed; 1: machine operator, hospitality staff, assistant, laborer, and related worker; 2: tradesperson, clerk and skilled officer, and sales and service staff; 3: other business manager, arts/media/sportsperson, and associate professional; 4: senior manager in a large business organization, government administration, and qualified professional.

**Table 2 behavsci-13-00824-t002:** Correlations for the measures used in the study.

	1.	2.	3.	4.	5.	6.	7.	8.	9.	10.	11.
1. Age											
2. Number of Books	0.02										
3. Parental Occupation	0.05	0.66 **									
4. Speed of processing	−0.04	0.17 *	0.23 **								
5. Planning	−0.09	0.27 **	0.25 **	0.39 **							
6. Attention	0.07	0.18 **	0.19 **	0.38 **	0.43 **						
7. Simultaneous processing	−0.07	0.20 **	0.22 **	0.28 **	0.40 **	0.33 **					
8. Successive processing	−0.02	0.08	0.08	0.15 *	0.17 *	0.17 *	0.34 **				
9. Wordchains	−0.07	0.21 **	0.25 **	0.40 **	0.29 **	0.34 **	0.30 **	0.32 **			
10. CBM-Maze	0.00	0.20 **	0.26 **	0.58 **	0.40 **	0.41 **	0.35 **	0.28 **	0.65 **		
11. MAT ^a^	0.02	0.50 **	0.52 **	0.43 **	0.54 **	0.37 **	0.45 **	0.27 **	0.38 **	0.46 **	
12. MRT ^b^	0.04	0.46 **	0.45 **	0.33 **	0.48 **	0.27 **	0.42 **	0.22 **	0.28 **	0.36 **	0.74 **

Note: ^a^ Mathematics Achievement Test; ^b^ Mathematics Reasoning Test. * *p* < 0.05; ** *p* < 0.01.

**Table 3 behavsci-13-00824-t003:** Results of hierarchical regression analysis predicting students’ performance.

		Wordchains		CBM-Maze		MAT		MRT	
Step	Independent Variables	β	ΔR^2^	β	ΔR^2^	β	ΔR^2^	β	ΔR^2^
	Model 1								
1	Number of Books at Home	0.07 **	0.07 ***	0.05 **	0.07 ***	0.29 ***	0.31 ***	0.35 ***	0.25 ***
	Parental Occupation	0.20 **		0.22 **		0.33 **		0.22 **	
2	Speed of processing	0.36 ***	0.13 ***	0.56 ***	0.29 ***	0.33 ***	0.10 ***	0.23 ***	0.05 ***
3	Planning	0.12 *	0.01 **	0.19 ***	0.03 ***	0.35 ***	0.10 ***	0.33 ***	0.09 ***
3	Attention	0.20 **	0.03 ***	0.21 ***	0.04 ***	0.17 **	0.02 ***	0.11	0.01
3	Simultaneous processing	0.18 **	0.03 ***	0.18 ***	0.03 ***	0.28 ***	0.07 ***	0.28 ***	0.07 ***
3	Successive processing	0.26 ***	0.07 ***	0.19 ***	0.03 ***	0.20 ***	0.04 ***	0.17 **	0.03 ***
	Model 2								
3	Planning	0.03	0.09 ***	0.10	0.08 ***	0.27 ***	0.14 ***	0.26 ***	0.13 ***
	Attention	0.14 *		0.14 *		0.04		0.02	
	Simultaneous processing	0.06		0.07		0.14 **		0.18 **	
	Successive processing	0.22 **		0.13 *		0.09		0.09	

Note: * *p* < 0.05; ** *p* < 0.01; *** *p* < 0.001.

## Data Availability

The data can be share upon request from the corresponding author.

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
