# Peer review of "Domain Specific and Cross Domain Associations between PASS Cognitive Processes and Academic Achievement"

_behavsci, 2023, doi:10.3390/bs13100824_

Round 1
Reviewer 1 Report
Check thoroughly for spelling errors (e.g. processses, spartial, dichotomazing).
Overall, this manuscript is well-written, with appropriate analyses. However, there are some theoretical gaps and some gaps in reporting of the results that I have suggested improvements for below. Additionally, I am requesting minor exploratory analyses to more clearly understand the role of PASS variables in predicting achievement.
-----------
Abstract: The authors conclude by saying "This further suggests that operationalizing intelligence in terms of four separate but interrelated neurocognitive processses might be instructionally more valuable than dichotomazing intelligence into fluid and crystallized or verbal and non-verbal." However, this point is not discussed further in the manuscript, and is not considered in the hypotheses. Please either remove or integrate throughout the manuscript.
Intro:
I'm curious as to why the authors didn't include Attention in their hypotheses. The literature review suggested attention has played a role in previous studies predicting math and reading, and there are several studies suggesting overlap between etiological factors influencing both attention with reading and attention with math. If the authors have a strong rationale for excluding attention, I ask that they provide this in the introduction.
The authors have not fully substantiated their choice of cognitive/executive functioning variables in the introduction. While they have provided useful information on PASS variables and the theory tying them to achievement, the authors have not explained why these variables over other executive functioning measures and operationalizations where chosen. While I do understand the reasoning behind using more objective measures of cognition as stated with reference to vocabulary subtests, I also think that other cognitive batteries exist that provide similar information to PASS. I am not disagreeing with using the PASS, but think the paper is currently missing a connection between the broader literature on cognitive/executive functioning operationalization and the current measures. Please provide these details to flesh out the rationale of the study. For example, processing speed is often included in a battery of executive functioning tasks rather than considered separately. Can the authors also provide their rationale for separating the cognitive skills more clearly?
Methods:
Please supply more detail on the timing of the assessments. How many days apart were the Phases? For all participants, was the order the same?
Results:
Terms for the measures in the intro don't match the terms used in the results. Please revise so that there is a one-to-one correspondence (e.g. aligning Visual Matching with Processing Speed). I suggest using the constructs (Planning, Attention, Processing Speed, etc.) from the intro in the written and tabled results, with notes in the tables to align the specific measure name to the construct, or including the construct with the measure label (e.g. Visual Matching - Processing Speed) in reporting.
Since the analyses within this study are correlational, please avoid causal terms such as 'significant effect' and instead use terms such as 'significant association' or 'significant influence.'
p. 6 & 7 "In Model 1, the four PASS processes collectively accounted for an additional 17% (Wordchains) and 23% (CBM-Maze) of the unexplained variance." - do the authors mean the variance unexplained by SES? This phrasing is somewhat confusing because if the variance was accounted for by the PASS variables, it wouldn't be unexplained in the model. Unexplained variance is generally considered what is left over after all predictors have been included. Please revise your wording for clarity.
Did the authors examine the change in R squared for significance? It would be interesting to note in Table 3. Additionally, because of the differential predictions of specific PASS skills for reading and math, it would provide useful information to the study if the authors conducted sensitivity/exploratory analyses where each PASS variable was entered as an independent 'block' to assess whether attention and successive processing provide the only significant changes in R squared for reading, and planning and simultaneous processing for math. If the model changes indicate that planning and simultaneous processing do not contribute significantly to variance in reading, for example, then they may be dropped from the models. However, even though they are not independently significant predictors, their contribution to the change in R squared may be large enough to warrant keeping them in the models. This distinction can help tease apart their importance in predictive models of achievement.
Discussion:
I do not find it surprising that attention was a significant predictor of reading, and think the authors could devote some space in the discussion to the extant literature associating attentional skills and ADHD with reading outcomes.
The number of self-citations is larger than usually seen. The discussion could also benefit from comparing and contrasting the literature that examines similar executive functioning skills to achievement outcomes. There are multiple other theoretical approaches and measures that have examined the association between cognitive/executive functioning and achievement that should be referred to (at least collectively) when interpreting the current results. Overall, bringing the results into line with the broader literature on cognitive/executive functioning skills and achievement will broaden the scope of the paper, which is currently limited to one theory and one operationalization of cognition.
There are several spelling mistakes and a few grammatical mistakes that should be fixed.
Reviewer 2 Report
1. Clear and understandable abstract which motivates the reader to continue reading the manuscript
2. There are too many references on the work of the authors (more than 10). This phenomenon weakens the present study and its contribution.
3. "Results of hierarchical....mathematics"
Rephrase
4. in my opinion the terms mathematics and arithmetic cannot be used as synonymous. Some of the studies which are referred do not examine mathematics (e.g. the references which are used for the first sentence)
5. "Number sense...for mathematics"
in the case of language cognitive skills are discussed. In the case of mathematics, number sense and counting are not cognitive skills. Cognitive psychologists examined memory, processing efficiency and so on, as relevant cognitive skills. The specific section has to be enriched.
6. based on the given information:
273 families (not parents) was the population and 242 was the sample? (extremely high percentage of participation which is probably unique). Which was the number of students who were excluded due to an immigration background? And what do you mean with this background (how is this related with the knowledge of the greek language etc)
7. information about the mothers' and fathers' occupations were used. Today there are too many divorced parents. Explain how the parents' occupations are used in order to have the SES of the families.
8. Too many tests are used. At the presentation of those tests we need to know whether there were constructed and used for specific ages in order to be able to judge their suitability for the 6th graders.
9. excellent presentation of the results
10. I believe that limitations have to be presented at a different section and there are methodological limitations which have to be added.
Round 2
Reviewer 1 Report
Please see attached file.

Minor edits needed.
Reviewer 2 Report
Thank you for the changes that have been done.
Author Response
Thank you
Round 3
Reviewer 1 Report
The role of PASS processes specifically, and in relation to other executive functioning/intelligence measures is still not well articulated within the article. Using the rationale of “this is not the focus of this study” is not sufficient. Focusing on PASS solely is not justified by the context presented. Limiting operationalization of a construct to one measure is a significant weakness of this study (e.g. low content validity and restricted generalization). Using several measures to create a latent variable is one way to strengthen the construct validity of a research study, but I understand that this is not always possible. However, researchers can mitigate this limitation by aligning the available measures to the broader available research and explicitly stating similarities and differences that exist.
RESPONSE: We agree with the reviewer that using one measure to operationalize the PASS processes is a weakness of the study and this is the reason we include it in the limitations section. We also mention the connection between the CAS measures and measures from the broader literature (see Methods section). Finally, we draw connections to the broader literature in the Discussion section (see page 8).
In the discussion, please provide a further comparison and contrast of the current results with other results using similar EF/Intelligence measures beyond Digit Span forward (e.g. other measures of planning, attention, and simultaneous processing).
minor edits can be fixed with an additional read through of the document.
Author Response
We appreciate your feedback. We compare the current findings with other studies that use similar EF/Intelligence measurements (see page 8).